PREPARED FOR SUBMISSION TO JHEP

CERN-TH-2024-193

# Locality and Conserved Charges in $T\overline{T}$-Deformed CFTs

**Ruben Monten,**[a] **Richard M. Myers,**[b] **Konstantinos Roumpedakis**[c]

[a] *Theoretical Physics Department, CERN, 1211 Geneva 23, Switzerland*

[b] *Mani L. Bhaumik Institute for Theoretical Physics*
*Department of Physics & Astronomy, University of California, Los Angeles, CA 90095, USA*

[c] *William H. Miller III Department of Physics and Astronomy, Johns Hopkins University,*
*3400 North Charles Street, Baltimore, MD 21218, U.S.A.*

*E-mail:* ruben.monten@cern.ch, myersr@physics.ucla.edu, kroumpe1@jh.edu

ABSTRACT: We investigate the locality properties of $T\overline{T}$-deformed CFTs within perturbation theory. Up to third order in the deformation parameter, we find a Hamiltonian operator which solves the flow equation, reproduces the Zamolodchikov energy spectrum, and is consistent with quasi-locality of the theory. This Hamiltonian includes terms proportional to the central charge which have not appeared before and which are necessary to reproduce the correct spectrum. We show that the Hamiltonian is not uniquely defined since it contains free parameters, starting at second order, which do not spoil the above properties. We then use it to determine the full conserved stress tensor. In our approach, the KdV charges are automatically conserved to all orders but are not a priori local. Nevertheless, we show that they can be made local to first order. Our techniques allow us to further comment on the space of Hamiltonians constructed from products of KdV charges which also flow to local charges in the deformed theory in the IR.

## 1   Introduction and Summary

Zamolodchikov's composite $T\overline{T}$ operator [1] generates a deformation of 2d QFTs which is irrelevant but solvable [2–4]. This means that certain quantities, such as the S-matrix, the partition function, and the energy spectrum, are known explicitly in terms of the seed theory. These results have impacted several fields in theoretical physics: they have been investigated from the point of view of effective and non-critical string theory [2, 5–11], they have broadened our understanding of integrable theories [3, 4, 12–17], and they are related to deformations of AdS/CFT [18–27] as well as holography beyond AdS [28–33].

Since the $T\overline{T}$ deformation is irrelevant, it allows us to explore a class of theories that flow to a given (undeformed) $\mathrm{QFT_{IR}}$ in the IR. In the UV, on the other hand, these theories differ radically from conventional field theories defined within the Wilsonian paradigm. For instance, based on properties of the deformed correlation functions, symmetry generators, and the S-matrix, it has been argued that the theory does not respect locality at scales smaller than the one set by the dimensionful deformation parameter [34–39]. We will refer to theories of this type, that are only local at sufficiently large distances, as "quasi-local". The subject of this work is to investigate how $T\overline{T}$-deformed theories are organized, by leveraging the structure preserved by the deformation to investigate their locality properties.

We restrict our attention to the deformation of two-dimensional CFTs. More specifically, we consider the $T\overline{T}$ deformation perturbatively in the canonical operator formalism.

We calculate the Hamiltonian and the stress tensor (including their off-diagonal components) up to third order in the deformation parameter, and the KdV charges up to first non-trivial order. We express them as functions of the free Virasoro generators of the undeformed CFT. As we will explain in detail, this requires a careful account of ordering ambiguities and regularization.

We take as the definition of the $T\overline{T}$ deformation that it relates a family of actions, parameterized by $\lambda$, through the flow equation

$$\partial_\lambda S = \frac{1}{2\pi} \int d^2x \, \det T_\lambda \,, \tag{1.1}$$

where $T_\lambda$ is the energy momentum tensor of the deformed theory and $S|_{\lambda=0}$ is the action of the Euclidean seed CFT. As a composite operator, $\det T_\lambda$ naively contains divergences from operator collisions and is only completely defined after regulation and subtraction. However, Zamolodchikov [1] showed that these divergences are not without structure. Instead they always take the form of total derivatives, implying that some quantities have a "universal" deformation in the sense that they can be computed without reference to how $\det T_\lambda$ has been regulated.

In particular, (1.1) leads unambiguously to an equation for the energy spectrum [3, 4], the solution of which is given by the square root formula

$$E_\lambda = \frac{1}{2\lambda}\Big(\sqrt{1 + 4\lambda E_0 - 4\lambda^2 P^2} - 1\Big), \tag{1.2}$$

where $E_0$ and $P$ are the energy and momentum eigenvalues of the undeformed theory. Other quantities however, such as the full Hamiltonian operator, are not necessarily universal in the above sense: they may depend on the details of the regularization and the total derivatives added to $\det T_\lambda$. We take an a priori agnostic stance and find indications that, for example, the stress tensor is non-universal. One can gain some intuition for these issues by noting that, classically, eq. (1.1) implies the equations

$$\partial_\lambda \mathcal{H}_\lambda = \det T_\lambda + (\text{total derivatives}) \,, \qquad \partial_\lambda \mathcal{P}_\lambda = (\text{total derivatives}) \,, \tag{1.3}$$

for the Hamiltonian and momentum densities. Assuming that they continue to hold quantum mechanically,[1] the total time derivatives, meaning commutators with the deformed Hamiltonian, will affect the integrated Hamiltonian (although they leave the spectrum invariant). There may therefore be a non-trivial family of Hamiltonians that provide $T\overline{T}$ deformations of the same theory, in the sense that the action satisfies eq. (1.1), corresponding to different regularized definitions of the deforming operator.[2]

The explicit regularization scheme we will use is to convolve local operators with a sufficiently smooth "smearing" function so that all singularities are regularized. We lay out

---

[1]Later, in section 2.3, we check that these equations are indeed satisfied at the quantum level.

[2]When renormalizing composite operators, a difference in scheme can produce a different operator. For example, in the $bc$ system one could define the operator $(TT)(\sigma)$ by normal ordering the mode operators, $:TT:$, or by subtracting Wick contractions of the 2-point functions. Though both will define a finite local operator, matrix elements of these operators will differ; see e.g. exercise 2.13(a) of [40].

our conventions in appendix A. As we will see, this regularization of operators allows us to elegantly parameterize the ambiguity in how the composite $T\overline{T}$ operator is defined, and it preserves the structure we use to organize our calculations.[3]

Our goal is to construct the deformed Hamiltonian operator within perturbation theory. This means that we work within the Hilbert space of the seed CFT and seek to construct an operator on that Hilbert space which, at each order in $\lambda$, is local, Hermitian, and reproduces the spectrum (1.2). We will express all operators in terms of the Virasoro modes of the undeformed CFT.[4] To systematically construct the most general Hamiltonian operator, we first introduce an auxiliary operator,

$$\tilde{H}_\lambda = \frac{1}{2\lambda} \left( \sqrt{1 + 4\lambda H_0 - 4\lambda^2 P^2} - 1 \right) , \tag{1.4}$$

which we dub the "fake" Hamiltonian. Here $H_0$ and $P$ are the Hamiltonian and momentum of the seed CFT. This operator is clearly non-local, in the sense that it is not the integral of a local density. However, it is a well-defined operator[5] which does not suffer from any ordering ambiguities or contact divergences, and it has the spectrum (1.2).

The true, (quasi)-local Hamiltonian $H_\lambda$ of the deformed theory must be related to the fake Hamiltonian $\tilde{H}_\lambda$ by a unitary transformation in order to have the spectrum (1.2). In other words, there must exist some anti-Hermitian operator $X_\lambda$ such that

$$H_\lambda = e^{-\lambda X_\lambda} \tilde{H}_\lambda e^{\lambda X_\lambda} . \tag{1.5}$$

The operator $e^{-\lambda X_\lambda}$ maps the energy and momentum eigenstates of the seed CFT to the energy and momentum eigenstates of the deformed Hamiltonian $H_\lambda$.[6] The requirement of unitarity implies that the latter remain orthonormal.

We construct $X_\lambda$ order by order in $\lambda$ by demanding $H_\lambda$ commutes with the undeformed momentum[7] and that it is local at each order in perturbation theory. The latter condition is consistent with both exact and quasi-locality of the full Hamiltonian. The distinction between the two is that the number of derivatives appearing in the Hamiltonian of a truly local theory is globally capped, whereas their number increases with each order in perturbation theory for quasi-local theories. There exists a different type of non-locality, which is directly visible in perturbation theory. For example, the operator $X_\lambda$ itself cannot be a local operator since it maps the non-local operator $\tilde{H}_\lambda$ to the local operator $H_\lambda$. The operator $e^{-\lambda X_\lambda}$ generates the flow of states $|n\rangle_\lambda = e^{-\lambda X_\lambda} |n\rangle$ and is the integral of a local operator over a full Cauchy slice [41, 42]. This is not in contradiction with "locality" of

---

[3]Although we will obtain regularized expressions, they do not necessarily remain finite when we take the regulator away. To make them finite, we would need to add counterterms and subtract the divergences. We will not work out the details of this renormalization procedure here.

[4]This should imply no loss of generality, since the seed Hamiltonian $H = L_0 + \overline{L}_0$ and the deformation operator are also constructed only from the seed theory's Virasoro modes.

[5]This operator is not Hermitian for $\lambda < 0$ but perturbation theory is not sensitive to this.

[6]As such, $X_\lambda$ is related to the generator of the flow of states in [41, 42]. They coincide at lowest order in $\lambda$. Furthermore, the operator $X_\lambda$ can be seen as the quantum equivalent of generator of a classical canonical transformation. For a related discussion, see [43].

[7]This is the assumption that, even quantum mechanically, the $T\overline{T}$ flow leaves the momentum operator invariant.

the theory, which is determined by the locality of the Hamiltonian operator, not by any notion of locality in the operators needed to produce states from the vacuum.[8] Part of the motivation for this work is to determine whether the Hamiltonian and KdV charges are free of this type of non-locality, and are indeed quasi-local.

We find that, while the space of unitary transformations (1.5) is naively large, the requirement of locality is severely restrictive. Up to third order in $\lambda$, beside a choice in regulation scheme, there is a 1-parameter family of local Hamiltonians with the spectrum (1.2). This family is compatible with an ordering of the Hamiltonian

$$H_\lambda = \int \frac{d\sigma}{2\pi} \frac{1}{2\lambda} \left( \sqrt{1 + 4\lambda\mathcal{H}_0 - 4\lambda^2\mathcal{P}_0^2} - 1 \right) + \mathcal{O}(c), \tag{1.6}$$

where $\mathcal{O}(c)$ indicates terms proportional to the central charge. In the special case where $c = 0$ and the seed CFT is the free boson, this Hamiltonian coincides with Nambu-Goto theory, as was already argued in [4, 18, 43].[9] However, we find that when $c \neq 0$ there are new terms which are essential and cannot be removed by a change in regularization scheme. When the full Hamiltonian is written as $H_\lambda = H_0 + \lambda H_1 + \lambda^2 H_2 + \cdots$ our result in terms of the undeformed Virasoro modes is given in eqs. (2.13), (2.19) and (2.23). As a cross check, we compute directly the deformed energy spectrum of an arbitrary primary state in section 2.2.

Given our family of Hamiltonians, we proceed to check whether there exists a conserved stress tensor such that the integral of $T_{tt}$ yields the Hamiltonian, and whether such a stress tensor satisfies the $T\overline{T}$ flow equation. The answer to both questions is affirmative; (1.3) can be satisfied for some choice of the total derivative terms. We find that the choice of total derivatives in (1.3) depends on the same coefficients parametrizing the family of Hamiltonians obtained from (1.5). This step therefore enables us to link the ambiguity in the deformed Hamiltonian to an ambiguity in the definition of the $\det T_\lambda$ operator.

The calculation of the stress tensor serves not only as a cross-check of our results, but also gives us access to the Hamiltonian and momentum density operators, which contain more information than the integrated quantities. In particular, we can check the validity of the "$T\overline{T}$ trace equation"

$$\text{Tr}\, T + 2\lambda \det T = 0 . \tag{1.7}$$

We show that this equation can indeed be satisfied up to second order for certain additional choices of the total derivative ambiguities.

The process of constructing a conserved stress tensor that satisfies the flow equations (1.3) could have been performed independent of the construction (1.5). Remarkably, the space of Hamiltonians obtained using conservation coincides exactly with the results of the unitary transformation. However, the result of this method is an expression for the full stress

---

[8]Indeed this is the case even in scattering where creation/annihilation operators are related to the local fields by a Fourier transform. This situation is also familiar from three-dimensional gauge theories [44] where monopole operators cannot be expressed in a local way using the gauge field.

[9]Even when related to more general, non-critical string theory, the total central charge including contributions from the ghosts and Liouville term adds up to 0 [10].

tensor, not just the integrated Hamiltonian, so it is more complicated and requires a larger set of coefficients to parameterize the total derivative ambiguities. We can interpret some of these as the freedom to add $\lambda$-dependent improvement terms to the stress tensor, while others correspond to genuine ambiguities that already appear in the unitary transformation.

With the unitary transformation $e^{-\lambda X_\lambda}$ mapping the energy and momentum eigenstates of the undeformed CFT to those of the deformed theory, we can ask how it acts on the KdV charges $I_0^{(k)}$. We define the "conjugated KdV charges"[10]

$$\hat{I}^{(k)} \equiv e^{-\lambda X_\lambda} I_0^{(k)} e^{\lambda X_\lambda} \ , \tag{1.8}$$

which are generically non-local. They have the same spectrum as the undeformed charges, rather than the predicted $T\overline{T}$-deformed spectrum [12, 42]. It is clear that the conjugated charges all remain conserved and pairwise commuting. At least to low order in perturbation theory, we show that it is possible to construct functions of the $\hat{I}^{(k)}$ which are (quasi)-local and have the desired spectrum.

It is interesting to consider more general deformations to which these techniques can be applied; to this end we study the most general class of theories for which the deformed spectrum depends on the undeformed eigenvalues of the KdV charges. We consider generalized fake Hamiltonians, constrained only by dimensional analysis. This class of deformations still involves only the Virasoro models of the undeformed theory, and hence the deformed energy spectrum will depend only on the stress tensor sector of the theory. With these more general fake Hamiltonians, we again analyze whether a unitary transformation exists such that the new $H_\lambda$ in (1.5) is a (quasi)-local charge. We find that locality imposes restrictions on the allowed deformations.

The rest of this work is organized as follows. In section 2 we set the stage for our approach. We calculate $X_\lambda$ and $H_\lambda$ to third order in $\lambda$ in section 2.1. In section 2.2 we perform a direct check of our result by computing the perturbative spectrum of $H_\lambda$ on a primary state in the undeformed CFT. Finally, in section 2.3 we construct a conserved stress tensor such that $T_{tt}$ reproduces $H_\lambda$ and check that it satisfies the flow equations (1.3), allowing us to directly relate the ambiguity in the deformed Hamiltonian $H_\lambda$ to an ambiguity in the definition of the renormalized $\det T_\lambda$ operator. We additionally check the trace equation (1.7) and find that it holds to $\mathcal{O}(\lambda^2)$.

In section 3 we study the infinite tower of KdV charges. We show that to leading order in $\lambda$ one can find functions of the conjugated charges which are local and have the expected spectrum.

Finally, in section 4 we consider fake Hamiltonians which are general functions of the KdV charges and ask which of these Hamiltonians can be related to a local Hamiltonian by a unitary transformation. We find that locality presents a strong constraint on the space of possible deformations.

---

[10]These "conjugated KdV charges" are not to be confused with the higher KdV analogs of the fake Hamiltonian (1.4). We note that these conjugated KdV charges agree at first order with the "flowed charges" of [42], but they differ at higher orders due to the $\lambda$-dependence of $X_\lambda$.

## 2  Locality, spectrum and conservation

We will consider the $T\overline{T}$ deformation of a Euclidean 2d seed CFT on the cylinder with stress tensor components $T$ and $\overline{T}$, from which we may define the Virasoro mode operators

$$T(\sigma) = \sum_{n\in\mathbb{Z}} L_n e^{in\sigma}, \quad \overline{T}(\sigma) = \sum_{n\in\mathbb{Z}} \overline{L}_n e^{-in\sigma} , \tag{2.1}$$

where $\sigma \sim \sigma + 2\pi$ is the angular direction on the cylinder. They obey the algebra[11]

$$[L_m, L_n] = (m-n)L_{m+n} + \frac{c}{12}m^3\delta_{m+n}. \tag{2.2}$$

With these definitions, $H_0 = L_0 + \overline{L}_0$ and $P = i(L_0 - \overline{L}_0)$ are the Hamiltonian and momentum of the seed CFT. We assume they are local, by which we mean that there exist densities $\mathcal{H}_0(\sigma)$ and $\mathcal{P}_0(\sigma)$ which are local functionals of the fundamental fields in the theory, such that

$$H_0 = \int \frac{d\sigma}{2\pi}\mathcal{H}_0(\sigma), \quad P = \int \frac{d\sigma}{2\pi}\mathcal{P}_0(\sigma) . \tag{2.3}$$

Since the $T\overline{T}$ deformation can be written purely in terms of the stress tensor, we use these ingredients to analyze the stress tensor sector of the deformed theory, independent of the specific CFT we started with.

Throughout this work, dimensional analysis plays an important role in restricting the operators allowed to appear at each order in perturbation theory. Our seed theory is a CFT on the cylinder, where the only dimensionful scale is the circumference, $2\pi R$, though we have set $R$ to unity to simplify expressions. Naively, since the integrand of (1.1) does not depend on $R$, the circumference is also not expected to enter the deformation of the Hamiltonian and momentum densities.[12] Noting the length dimensions $[\lambda] = 2$, $[T] = [\overline{T}] = -2$, the requirement that the Hamiltonian density[13] have dimension $[\mathcal{H}_\lambda] = -2$ then places severe restrictions on the combinations of $T, \overline{T}$, and their derivatives that can appear at each order: at each order in $\lambda$ the $T\overline{T}$-deformed Hamiltonian is a polynomial in stress tensor components of the seed CFT.[14]

We have been careful to indicate that the above argument is the naive conclusion because $\det T_\lambda$ is a composite operator and must be renormalized. Regulators generally

---

[11]The more familiar central extension $\frac{c}{12}m(m^2 - 1)$ on the plane is related by a shift in $L_0$. We comment on the importance of this shift in Section 2.1.4.

[12]Results based on the flow equation for semi-classical operators in [42] may alter this expectation. However, in this paper we use the absence of $R$ in the densities as a minimal assumption, and find that a solution indeed exists. It would be interesting to compare the results we obtain here more directly with the semi-classical flow equation.

[13]Throughout this section we do dimensional analysis at the level of the densities rather than the integrated quantities for convenience. Working with the integrated quantities would involve giving summations an effective dimension.

[14]Since we work within perturbation theory, we do not comment on the possible resummation of the Hamiltonian into some local, but non-polynomial, functional. In the special case $c = 0$, this resummation is known to occur classically [4].

include a scale, the simplest example of this is point splitting where the splitting distance can be turned into a dimensionless parameter by introducing factors of $R$. Throughout this work we use versions of operator smearing, as reviewed in appendix A. But as we will see, since this $R$-dependence can enter only through the regulator, the violation of our dimensional analysis can be controlled.

## 2.1   Unitary Transformation

While the fake Hamiltonian (1.4) is well-defined, e.g. it contains no ordering issues, it is non-local since it is a non-linear functional of integrated quantities. It is also diagonalized by the same basis of energy and momentum eigenstates as the seed theory, which is not the case for the $T\overline{T}$-deformed Hamiltonian [41–43].

A Hamiltonian with more desirable properties must be unitarily related to the fake Hamiltonian (1.4) since their spectra match.[15] The space of possible real Hamiltonians $H_\lambda$ describing the $T\overline{T}$ spectrum can then be spanned by the operators

$$H_\lambda = e^{-\lambda X_\lambda} \tilde{H}_\lambda e^{\lambda X_\lambda} \ , \tag{2.4}$$

where $X_\lambda$ is some anti-unitary operator. Since the $T\overline{T}$ deformation acts trivially on the undeformed momentum, we will also need to require that

$$P = e^{-\lambda X_\lambda} P e^{\lambda X_\lambda}, \tag{2.5}$$

implying $[P, X_\lambda] = 0$. It also follows that $[H_\lambda, P] = 0$.

Expanding functions of $\lambda$ as $F_\lambda = F_0 + \lambda F_1 + \lambda^2 F_2 + \lambda^3 F_3 + \cdots$ to third order in $\lambda$, equation (2.4) yields

$$
\begin{aligned}
H_0 - \tilde{H}_0 =&\, 0 \ , \\
H_1 - \tilde{H}_1 =&\, [\tilde{H}_0, X_0] \ , \\
H_2 - \tilde{H}_2 =&\, [\tilde{H}_0, X_1] + [\tilde{H}_1, X_0] + \frac{1}{2}\left[[\tilde{H}_0, X_0], X_0\right] \ , \\
H_3 - \tilde{H}_3 =&\, [\tilde{H}_0, X_2] + [\tilde{H}_2, X_0] + [\tilde{H}_1, X_1] + \frac{1}{2}\left[[\tilde{H}_1, X_0], X_0\right] \\
&+ \frac{1}{2}\left[[\tilde{H}_0, X_1], X_0\right] + \frac{1}{2}\left[[\tilde{H}_0, X_0], X_1\right] + \frac{1}{6}\left[\left[[\tilde{H}_0, X_0], X_0\right], X_0\right] .
\end{aligned}
\tag{2.6}
$$

In the remainder of this section we solve these equations order by order to determine $H_i$ and $X_i$.

Before we proceed, let us comment on an ambiguity in $X_\lambda$. We can modify the unitary transformation as

$$e^{-\lambda X_\lambda} \to e^{-\lambda X_\lambda} e^{-\lambda \mathcal{O}_\lambda}, \quad [\mathcal{O}_\lambda, \tilde{H}_\lambda] = 0 \ , \tag{2.7}$$

---

[15]To match spectra, we strictly only require the real and fake Hamiltonans to be related by a similarity transformation. However, it is always possible to demand that the eigenvectors of $H$ be orthonormal, so there is no loss in generality demanding the transformation be unitary.

and equation (2.4) remains unchanged. Any operator $\mathcal{O}_\lambda$, not necessary local, that commutes with both the Hamiltonian and momentum $H_0$ and $P$ has this property. In turn, this produces an ambiguity in the definition of $X_\lambda$

$$
\begin{aligned}
X_0 &\to X_0 + \mathcal{O}_0 \ , \\
X_1 &\to X_1 - \frac{1}{2}[X_0, \mathcal{O}_0] + \mathcal{O}_1 \ , \\
X_2 &\to X_2 - \frac{1}{2}[X_1, \mathcal{O}_0] + \frac{1}{2}[X_0, \mathcal{O}_1] + \frac{1}{12}[X_0, [X_0, \mathcal{O}_0]] + \frac{1}{12}[\mathcal{O}_0, [\mathcal{O}_0, X_0]] + \mathcal{O}_2 \ ,
\end{aligned}
\tag{2.8}
$$

and similarly at higher order.

### 2.1.1 First order

At first order, the calculation is straightforward, but we begin by treating it in detail to demonstrate our approach.

As described in the beginning of this section, if $H_1$ is to come from a local density, that density must have length dimension $-4$. Building operators out of $T$, $\overline{T}$, and their derivatives, it is clear that, schematically, the only operators which can appear take the form $T^2, T\overline{T}, \overline{T}^2$, and $T'', \overline{T}''$. The final two operators are total derivatives[16] and hence will not contribute to $H_1$. Using (2.1) to write all such operator contributions in terms of the Virasoro modes, we would obtain the general operator[17]

$$
H_1 = \sum_{n \in \mathbb{Z}} \left( g_{2,0}^{(1)} L_n L_{-n} + g_{1,1}^{(1)} L_n \overline{L}_n + g_{0,2}^{(1)} \overline{L}_n \overline{L}_{-n} \right).
\tag{2.9}
$$

Classically, (2.9) is the most general local charge built out of the Virasoro modes. Quantum mechanically, this object is plagued by UV divergences and ordering ambiguities. One might be tempted to try parameterizing the possible operator orderings, but this quickly becomes cumbersome at higher order and one has no guarantee that the resulting operator will be a finite, well-defined operator.

A more elegant, and computationally useful, solution is to introduce a smearing regulator, which we review in Appendix A. In terms of the Virasoro modes, this means we multiply our operators by products of Fourier modes of a smearing function[18] $w_n$ to write

$$
H_1 = \sum_{n \in \mathbb{Z}} \left( g_{2,0}^{(1)} L_n L_{-n} + g_{1,1}^{(1)} L_n \overline{L}_n + g_{0,2}^{(1)} \overline{L}_n \overline{L}_{-n} \right) w_n.
\tag{2.10}
$$

When doing this we necessarily involve higher derivative orders through $w_n$ than allowed for by the naive version of dimensional analysis described at the beginning of this section. However, as we noted there, arbitrary $R$-dependence is only allowed to enter via the regulator so there is no tension between those arguments and the use of a regulator here.

---

[16]Their contribution to the density can, however, be understood as improvement terms. Some properties, e.g. the trace equation, are sensitive to such contributions, we as will see in section 2.3.

[17]Note we also demand $[H_1, P] = 0$, as discussed previously.

[18]We require that $w_{-n} = w_n$ and $w_0 = 1$. Example regulators with these properties include the heat kernel $w_n = e^{-\epsilon|n|}$ and cutoff $w_n = \theta(|n| < N)$. The particular choice will not be important for our results.

An immediate advantage of explicitly including the regulator $w_n$ is that it gives us a way to think about the other operator orderings not explicitly represented in (2.10). Firstly, integrating a smeared product of stress tensor components will only ever produce a sum over all integers and a string of Virasoro modes, and never orderings in the sum such as $\sum_{m>n}$. Any other ordering operation will either reduce to a reindexing, which we are always allowed to do since our sums are regulated to be finite, such as $w_n L_n L_{-n} \to w_n L_{-n} L_n$, or should be regarded as equivalent to a change of scheme, and hence parameterized by the choice of regulator $w_n$.

With our demands on $H_\lambda$ now imposed at first order, we can ask whether there exists an operator $X_0$ solving the constraint (2.6). We can characterize, in general, when this type of problem has a solution. Since the constraints (2.6) are linear, we can investigate the generic problem

$$L_{n_1} \cdots L_{n_N} \overline{L}_{m_1} \cdots \overline{L}_{m_M} = [L_0 + \overline{L}_0, X_0]. \tag{2.11}$$

By inspection, this has solution

$$X_0 = -\frac{1}{n_1 + \cdots + n_N + m_1 + \cdots m_M} L_{n_1} \cdots L_{n_N} \overline{L}_{m_1} \cdots \overline{L}_{m_M} + \mathcal{O}_0 , \tag{2.12}$$

whenever $n_1 + \cdots + n_N + m_1 + \cdots + m_M \neq 0$ and for any operator $\mathcal{O}_0$ that commutes with both $L_0$ and $\overline{L}_0$, since $X_0$ is to commute with $P$. If $n_1 + \cdots + n_N + m_1 + \cdots + m_M = 0$, then a solution for $X_0$ does not exist.

Demanding existence of $X_0$ and (2.6) immediately implies $g_{2,0}^{(1)} = g_{0,2}^{(1)} = 0$ and $g_{1,1}^{(1)} = -4$ so the first order Hamiltonian is uniquely determined to be

$$H_1 = -4 \sum_{n \in \mathbb{Z}} L_n \overline{L}_n w_n , \tag{2.13}$$

and the solution for $X_0$ is[19]

$$X_0 = 2 \sum_{n \neq 0} \frac{1}{n} L_n \overline{L}_n w_n + \mathcal{O}_0. \tag{2.14}$$

We note that $X_0$ is non-local. This should be expected since it defines a map between a non-local operator, the fake Hamiltonian, and a local one, $H_1$. By inspection, the most general form of (2.6) is

$$\mathcal{O}_0 = i\alpha_{2,0} L_0 L_0 + i\alpha_{1,1} L_0 \overline{L}_0 + i\alpha_{0,2} \overline{L}_0 \overline{L}_0$$
$$+ \sum_{n \neq 0} \frac{1}{n} \left( f L_n L_{-n} + \overline{f} \overline{L}_n \overline{L}_{-n} \right) w_n. \tag{2.15}$$

Note that there is no requirement of locality in this expression, so products of $L_0$ and $\overline{L}_0$ are allowed. However, we do demand that $X_\lambda$ be anti-Hermitian, which requires all the undetermined constants to be real.

---

[19]We note again that this expression agrees with the flow of states operator $\mathcal{X}$ of [41, 42] at $\lambda = 0$.

### 2.1.2 Second order

At the next order there are more operators, but the basic logic is the same. The local density producing $H_2$ should have length dimension $-6$. The only operators constructed from the stress tensor can again be organized by derivative order, schematically

$$
\begin{aligned}
\partial^0 : \quad & T^3,\ T^2\overline{T},\ T\overline{T}^2,\ \overline{T}^3, \\
\partial^2 : \quad & T^2,\ T\overline{T},\ \overline{T}^2, \\
\partial^4 : \quad & T,\ \overline{T},
\end{aligned}
\tag{2.16}
$$

where at each derivative order we can sprinkle derivatives in any $T$ or $\overline{T}$. The terms with a single stress tensor factor are total derivatives and hence can be ignored. In terms of the Virasoro modes, this yields the general form

$$
\begin{aligned}
H_2 = \sum_{n\in Z} n^2 &\Big( g_{2,0}^{(2)} L_n L_{-n} + g_{1,1}^{(2)} L_n \overline{L}_n + g_{0,2}^{(2)} \overline{L}_n \overline{L}_{-n} \Big) w_n^2 \\
+ \sum_{m,n\in Z} &\Big( h_{3,0}^{(2)} L_m L_n L_{-m-n} + h_{2,1}^{(2)} L_m L_n \overline{L}_{n+m} \\
&+ h_{1,2}^{(2)} L_{m+n} \overline{L}_n \overline{L}_m + h_{0,3}^{(2)} \overline{L}_n \overline{L}_m \overline{L}_{-m-n} \Big) w_m w_n,
\end{aligned}
\tag{2.17}
$$

where the coefficients are real by Hermiticity. There is some ambiguity in how the regulating factors are added. The choice we use here is convenient for obtaining the cancellations in (2.6) required for $X_1$ to exist.[20]

Imposing (2.6) we find that a solution for $X_1$, and hence the unitary transformation, only exists when

$$
\begin{aligned}
h_{3,0}^{(2)} &= h_{0,3}^{(2)} = 0, \\
g_{2,0}^{(2)} &= g_{0,2}^{(2)} = \frac{c}{3}, \\
h_{1,2}^{(2)} &= h_{2,1}^{(2)} = 8.
\end{aligned}
\tag{2.18}
$$

This leaves the coefficient $g_{1,1}^{(2)}$ undetermined, so at this order there is a 1-parameter family of local Hamiltonians which have the expected $T\overline{T}$ spectrum. This correction to the Hamiltonian, $H_2$, is

$$
\begin{aligned}
H_2 = 8 \sum_{m,n\in\mathbb{Z}} &\Big( L_m L_n \overline{L}_{m+n} + L_{m+n} \overline{L}_n \overline{L}_m \Big) w_m w_n \\
+ \frac{c}{3} \sum_{n\in\mathbb{Z}} n^2 &\Big( L_n L_{-n} + \overline{L}_n \overline{L}_{-n} \Big) w_n^2 + g_{1,1}^{(2)} \sum_{n\in\mathbb{Z}} n^2 L_n \overline{L}_n w_n^2 .
\end{aligned}
\tag{2.19}
$$

---

[20]In the language of appendix A, at first order we used product smearing, while here at second order we use operator smearing. As discussed in the appendix, there is nothing inconsistent about this combination of choices. If one instead chose to use exclusively operator smearing, one would instead find that $X_1$ exists only if some non-local operators are added to $H_2$ which vanish as the regulator is removed. A similar issue is encountered in section 2.3, where the smeared stress tensor is only conserved modulo terms that vanish as the regulator is removed.

It is useful to note that no reordering of the $L_n$'s is capable of absorbing the terms proportional to $c$ or $g_{1,1}^{(2)}$, because they would not produce the factor of $n^2$. Within our setup, this implies that these terms cannot be absorbed into a redefinition of the regulator or any counterterm. One could also imagine shifting $L_0$ by a constant, but such a redefinition also cannot absorb these terms due to the $n^2$ factor.[21] We treat such shifts in $L_0$ in detail in section 2.1.4.

The coefficient $g_{1,1}^{(2)}$ is not determined by the requirement that the second order spectrum matches with the one predicted for $T\overline{T}$-deformed CFTs and, as we will see below, it is also consistent with conservation of the stress tensor. This suggests that there exist multiple theories which all can be seen as valid $T\overline{T}$ deformed versions of the original theory and that the deformed Hamiltonian is not universal in the aforementioned sense. Similar undetermined constants appear at higher orders and correspond to the freedom to compose $e^{-\lambda X_\lambda}$ with a local unitary transformation.

It is possible that $g_{1,1}^{(2)}$ gets determined at higher orders, i.e. that the existence of a unitarity transformation requires a particular value for this coefficient, but we did not find any such constraint. We will see in section 2.2 that $g_{1,1}^{(2)}$ does not change the energy spectrum, but it does appear in off-diagonal components of the Hamiltonian operator. For example, for the normalized state $|\psi\rangle \propto L_{-k}\bar{L}_{-k}|h, \bar{h}\rangle$ built upon a primary state $|h, \bar{h}\rangle$

$$\langle\psi|H_2|0\rangle = g_{1,1}^{(2)} k^2 \ . \tag{2.20}$$

It is interesting to compare the expression for $H_2$ to the Nambu-Goto Hamiltonian, which is the classical (and $c = 0$) result [4],

$$H_{NG} = \int \frac{d\sigma}{2\pi} \frac{1}{2\lambda}\left(\sqrt{1 + 4\lambda\mathcal{H}_0 - 4\lambda^2\mathcal{P}_0^2} - 1\right). \tag{2.21}$$

Expanding, we would find that $H_2^{NG}$ matches the Hamiltonian above in the special case $c = 0$ and $g_{1,1}^{(2)} = 0$ with a particular choice of a regulator. One can modify the Nambu-Goto Hamiltonian to reproduce the new terms by adding $\frac{\lambda^2}{4}g_{1,1}^{(2)}(\mathcal{H}_0'^2 + \mathcal{P}'^2) - \frac{c\lambda^2}{6}(\mathcal{H}_0'^2 - \mathcal{P}'^2)$. It would be interesting to understand how the terms proportional to $c$ impact the holographic interpretation of the $T\overline{T}$ deformation. It is not clear how these terms, expressed as a function of the undeformed stress tensor, relate to the calculations that have appeared in the holography literature, which are phrased in terms of the fundamental fields instead.

As for the unitary transformation (2.4) that maps the fake Hamiltonian into eq. (2.19), we find

---

[21]For the special case of the free scalar, one could consider reorderings of the oscillators $\phi_k$ inside $L_n = \sum_k \phi_{n-k}\phi_k$. However, these can only produce shifts of $L_0$.

$$X_1 = -\sum_{\substack{m,n\neq 0 \\ m+n\neq 0}} \frac{m+n}{mn} w_m w_n (L_m L_n \bar{L}_{m+n} + L_{m+n} \bar{L}_m \bar{L}_n)$$

$$- 4\sum_{m\neq 0} \frac{w_m}{m} \left(L_m \bar{L}_0 \bar{L}_m + L_m L_0 \bar{L}_m\right)$$

$$- \sum_{n\neq 0} \frac{n}{2} g_{1,1}^{(2)} L_n \bar{L}_n w_n - \frac{1}{2}[X_0, \mathcal{O}_0] + \mathcal{O}_1 \,, \tag{2.22}$$

for any $O_1$ which commutes with both $L_0$ and $\bar{L}_0$. One can easily see that the above expression is anti-Hermitian using $L_n^\dagger = L_{-n}$ and $\bar{L}_n^\dagger = \bar{L}_{-n}$ for any choice of anti-Hermitian $O_i$.

### 2.1.3 Third Order

One proceeds in a similar fashion to the next order, to determine $H_3$ and $X_2$ in (2.6). For the former we get

$$H_3 = -16 \sum_{m,n,p} w_m w_p w_{m+p} \left(L_m L_n L_p \bar{L}_{m+n+p} + 3L_m L_n \bar{L}_p \bar{L}_{m+n-p} + L_{m+n+p} \bar{L}_m \bar{L}_n \bar{L}_p\right)$$

$$- \frac{2c}{3} \sum_{m,n} w_m w_n w_{m+n}(m^2 + mn + n^2) \left(L_m L_n L_{-m-n} + 3L_m L_n \bar{L}_{m+n} + 3L_{m+n} \bar{L}_m \bar{L}_n + \bar{L}_m \bar{L}_n \bar{L}_{-m-n}\right)$$

$$+ \sum_{m,n} w_m w_n (m+n)^2 \left(g_{2,1}^{(3)} L_m L_n \bar{L}_{m+n} + g_{1,2}^{(3)} L_{m+n} \bar{L}_m \bar{L}_n\right) + g_{1,1}^{(3)} \sum_n w_n n^4 L_n \bar{L}_n$$

$$+ \epsilon_1 \sum_n w_n L_n \bar{L}_n + \epsilon_2 (L_0 + \bar{L}_0) + \epsilon_3 \,, \tag{2.23}$$

where $g_{1,2}^{(3)}$, $g_{2,1}^{(3)}$ and $g_{1,1}^{(3)}$ are free parameters while the terms proportional to $\epsilon_i$ are counterterms. They are given by

$$\epsilon_1 = -\frac{16}{3} \sum_{m,n} w_m w_n w_{m+n}(m^2 + mn + n^2) \,,$$

$$\epsilon_2 = -\frac{2c}{9} \sum_{m,n} w_m w_n w_{m+n}(m^2 + mn + n^2)^2 \,,$$

$$\epsilon_3 = -\frac{c^2}{108} \sum_{m,n} w_m w_n w_{m+n} \left((m^2 + mn + n^3)^2 - 3m^2 n^2 (n+m)^2\right) \,.$$

Similar counterterms appear in the expression for $X_2$, which we have omitted. Hence we see that the Hamiltonian (2.23) is the classical expression in (2.21) supplemented by terms proportional to the central charge and counterterms. As before, the Hamiltonian is not uniquely determined and labeled by three free parameters $g_{1,2}^{(3)}$, $g_{2,1}^{(3)}$ and $g_{1,1}^{(3)}$.

### 2.1.4 Seed Hamiltonian and the plane

Throughout this work, we assume that our seed CFT has the standard Hamiltonian on the cylinder. This means we take $H_0 = L_0 + \overline{L}_0$ where the Virasoro modes are defined, as in (2.2), to have central extension of the form $\frac{c}{12}m^3\delta_{m+n}$. One may wonder how important this assumption is to our results, particularly in light of the observation that the original argument [3, 4] for the deformed spectrum (1.2) applied only on the cylinder and not on the plane, where the central extension would take the shifted form $\frac{c}{12}(m^3 - m)\delta_{m+n}$.

To make this precise, we phrase the question in terms of modifying the seed Hamiltonian to

$$H_0 = L_0 + \overline{L}_0 + \frac{c}{12}q \ , \tag{2.24}$$

where $q$ is a real constant and the modes $L_n$ and $\overline{L}_n$ are still the ones appropriate to the cylinder, obeying (2.2).

Alternatively, we could define the shifted modes $L'_n$ and $\overline{L}'_n$ by $L'_n = L_n$ for $n \neq 0$ and $L'_0 = L_0 + \frac{c}{24}q$, and similarly for the barred sector, so that

$$H_0 = L'_0 + \overline{L}'_0 \ . \tag{2.25}$$

These modes obey the Virasoro with a different form of the central extension,

$$[L'_m, L'_n] = (m - n)L'_{m+n} + \frac{c}{12}m(m^2 - q)\delta_{m+n}. \tag{2.26}$$

The special case $q = 1$ would be equivalent to taking the Hamiltonian (2.25) on the plane. It is important to note that on general grounds one would expect the $T\overline{T}$ flow to be sensitive to this type of change since the flow depends non-linearly on the seed theory, so this modification is not necessarily trivial.[22]

Our previous arguments proceed in largely the same manner, with the exception that there are new operators we can add to our local Hamiltonians $H_1$ and $H_2$, because $q$ carries units of energy.[23] At first order, we can add the operators

$$q^2, \ qL'_0, \ q\overline{L}'_0. \tag{2.27}$$

However, it is straightforward to check that (2.6) sets the coefficients of all these operators to zero at first order, so locality is insensitive to the change in the seed Hamiltonian at this order.

At second order the new operators are

$$q^3, \ q^2L'_0, \ q^2\overline{L}'_0, \ q\sum L'_n L'_{-n}, \ q\sum L'_n \overline{L}'_n, \ q\sum \overline{L}'_n \overline{L}'_{-n}. \tag{2.28}$$

---

[22]It would, however, be trivial if we selected $H_0 = L_0 + \overline{L}_0$ and instead wrote it as $H_0 = L'_0 + \overline{L}'_0 - \frac{c}{12}q$. Clearly, the seed Hamiltonian is unchanged and this amounts to a local "field" redefinition. A local redefinition will not change our conclusions about locality, and this can be checked directly (though it makes intermediate steps messier).

[23]Recall that we have chosen the dimensions of the Virasoro modes such that $c$ is dimensionless.

But now we find that a solution does not exist when we try to impose (2.6) – after exhausting all of our freedom to fix the free coefficients we are left with

$$H_2 - \tilde{H}_2 - \left( [\tilde{H}_0, X_1] + [\tilde{H}_1, X_0] + \frac{1}{2}[[\tilde{H}_0, X_0], X_0] \right) = \frac{qc}{3}(L_0'^{\,2} + \overline{L}_0'^{\,2}). \qquad (2.29)$$

Since the operator on the right commutes with $\tilde{H}_0$, it cannot be absorbed into a contribution to $X_1$. Since it is not local, it cannot be absorbed into our ansatz for $H_2$. Hence we conclude that there is no local Hamiltonian with the spectrum (1.2) unless either $c = 0$ or $q = 0$.

Motivated by these restrictions, one can further ask more generally about what class of fake Hamiltonia $\tilde{H}_\lambda$ are unitarily equivalent to a (quasi)-local Hamiltonian. We take up this question in section 4.

## 2.2 Perturbative Energy Spectrum

As a cross-check of our results, we can directly compute the perturbative energy spectrum of $H_\lambda = H_0 + \lambda H_1 + \lambda^2 H_2 + \cdots$. Since we started from the fake Hamiltonian and applied a unitary transformation, the expected $T\overline{T}$-deformed energy spectrum is guaranteed to come out by construction. However, we verify this result explicitly to illustrate that the terms proportional to $c$ are necessary to obtaining the correct spectrum, implying that these deviations from the Nambu-Goto Hamiltonian are necessary in the presence of non-zero central charge. In general, this would be a complicated problem in degenerate perturbation theory. But if we restrict to the special case of a primary state $|h, \overline{h}\rangle$, we shall see that the problem at second order, where a sum over degenerate states would first enter, simplifies significantly.

At first order we compute

$$E_{h,\overline{h}}^{(1)} = \langle h, \overline{h}|H_1|h, \overline{h}\rangle = -4h\overline{h} , \qquad (2.30)$$

which matches the first order expansion of (1.2).

At second order, we first note that all states outside the Verma module constructed atop $|h, \overline{h}\rangle$ are orthogonal to it. Within the Verma module, we can choose the states $|h, \overline{h}\rangle, L_n|h, \overline{h}\rangle, \overline{L}_n|h, \overline{h}\rangle$, and $L_n\overline{L}_m|h, \overline{h}\rangle$ to be orthogonal to all further descendent states, which we denote by $|N, \overline{N}\rangle$. Then since

$$\langle N, \overline{N}|H_1|h, \overline{h}\rangle = -4\sum_{n\in\mathbb{Z}} w_n \langle N, \overline{N}|(L_n\overline{L}_n|h, \overline{h}\rangle) = 0 , \qquad (2.31)$$

vanishes by orthogonality, $H_1$ has no matrix elements connecting the primary to the further descendent states $|N, \overline{N}\rangle$. Indeed, the same is true for the states $L_n|h, \overline{h}\rangle, \overline{L}_n|h, \overline{h}\rangle$ and $L_n\overline{L}_m|h, \overline{h}\rangle$ with $n \neq m$. Hence, to second order perturbation theory it is sufficient to sum only over intermediate states of the form $N_n L_n\overline{L}_n|h, \overline{h}\rangle$ where $N_n$ is a normalization factor[24]. With this, we compute

$$E_{h,\overline{h}}^{(2)} = \langle h, \overline{h}|H_2|h, \overline{h}\rangle + \sum_{n=1}^{\infty} \frac{|N_n|^2}{E_{h,\overline{h}}^{(0)} - E_{h+n,\overline{h}+n}^{(0)}} |\langle h, \overline{h}|L_n\overline{L}_n H_1|h, \overline{h}\rangle|^2. \qquad (2.32)$$

---

[24]It is straightforward to show $\frac{1}{N_n} = \sqrt{(2nh + \frac{c}{12}n^3)(2n\overline{h} + \frac{c}{12}n^3)}$.

It is straightforward to compute these terms and find

$$\langle h, \overline{h} | H_2 | h, \overline{h} \rangle = 8h\overline{h}(h + \overline{h}) + \frac{1}{18} \sum_{n=1}^{\infty} n(24h + cn^2)(24\overline{h} + cn^2)w_n^2$$

$$\sum_{n=1}^{\infty} \frac{|N_n|^2}{E_{h,\overline{h}}^{(0)} - E_{h+n,\overline{h}+n}^{(0)}} |\langle h, \overline{h} | L_n \overline{L}_n H_1 | h, \overline{h} \rangle|^2 = -\frac{1}{18} \sum_{n=1}^{\infty} n(24h + cn^2)(24\overline{h} + cn^2)w_n^2. \quad (2.33)$$

Thus, the correction to the energy spectrum is

$$E_{h,\overline{h}}^{(2)} = 8h\overline{h}(h + \overline{h}). \quad (2.34)$$

This result matches the $T\overline{T}$ spectrum (1.2). Importantly, the cancellation required to produce this answer occurs directly at the summand level, so we don't need to make explicit use of the regulator to obtain eq. (2.34).[25] This is a general feature we expect of the "universal" $T\overline{T}$-deformed observables, such as the energy spectrum, that they are independent of the regulation scheme.

## 2.3 Current Conservation

To get more fine-grained access to the $T\overline{T}$-deformed theory, we can calculate the full stress tensor densities that give rise to the (integrated) Hamiltonian we obtained above. The essential requirement we need to impose is the fact that it is conserved. We find that the resulting densities satisfy the flow equation eq. (1.3), for a particular choice of the total derivative terms.

In fact, as alluded to in the introduction, imposing the flow equation along with conservation of the stress tensor provides an independent way to derive the deformed Hamiltonian. This is the perspective we take in this subsection. It turns out that this approach leads to the same class of Hamiltonians as in Section 2.1.

The conservation equations $\dot{\mathcal{H}}_\lambda + \mathcal{P}_\lambda' = 0 = \dot{\mathcal{P}}_\lambda + \mathcal{K}_\lambda'$, where $\mathcal{K}_\lambda \equiv T_{\sigma\sigma}$, are compatible with the Virasoro algebra (2.2) in the undeformed CFT if we identify $\dot{\mathcal{O}} = -[H_\lambda, \mathcal{O}]$ and $\mathcal{O}' = -[P, \mathcal{O}]$. We parameterize the total derivatives in (1.3) by the functions $\mathcal{A}, \mathcal{B}, \mathcal{C}$, and $\mathcal{D}$ with prefactors chosen so that they are Hermitian,

$$\partial_\lambda \mathcal{H}_\lambda = \mathcal{H}_\lambda \mathcal{K}_\lambda - \mathcal{P}_\lambda^2 + \mathcal{A}' + i\dot{\mathcal{B}}, \qquad\qquad \partial_\lambda \mathcal{P}_\lambda = i\mathcal{C}' - \dot{\mathcal{D}}. \quad (2.35)$$

As before, we have been somewhat cavalier in writing the product of coincident local operators on the right-hand side of eq. (2.35), ignoring issues of ordering, divergences etc. We address these explicitly in perturbation theory by our smearing procedure, keeping in mind that the fundamental justification comes from Zamolodchikov's argument [1]. As explained in appendix B, this argument can be extended to all orders in perturbation theory.

Expanding eq. (2.35) perturbatively in $\lambda$, with $\mathcal{H}_\lambda = \mathcal{H}_0 + \lambda \mathcal{H}_1 + \ldots$ and similarly for the other operators, we find to first order that

$$\mathcal{H}_1 = \mathcal{H}_0 \mathcal{K}_0 - \mathcal{P}_0^2 + \mathcal{A}_0' - i[H_0, \mathcal{B}_0]. \quad (2.36)$$

---

[25]Note that this does not imply that we did not require the regulator implicitly as the manipulations used at intermediate steps are only justified when sums converge.

The most general ansatz which is compatible with locality, dimensional analysis, and which only depends on the stress tensor itself, is

$$\mathcal{A}_0 = \alpha_0 T' + \bar{\alpha}_0 \overline{T}' , \quad \mathcal{B}_0 = \beta_0 T' + \bar{\beta}_0 \overline{T}' , \quad \mathcal{C}_0 = \gamma_0 T' + \bar{\gamma}_0 \overline{T}' , \quad \mathcal{D}_0 = \epsilon_0 T' + \bar{\epsilon}_0 \overline{T}' . \quad (2.37)$$

Some combinations of the free coefficients $\alpha_0, \bar{\alpha}_0, \dots$ are determined by the conservation equation at first order

$$[H_0, \mathcal{H}_1] + [H_1, \mathcal{H}_0] + \mathcal{P}'_1 = 0 . \quad (2.38)$$

Using the Virasoro algebra, we find the constraints

$$\alpha_0 + \beta_0 - \gamma_0 - \epsilon_0 = \frac{c}{3} = \bar{\alpha}_0 - \bar{\beta}_0 + \bar{\gamma}_0 - \bar{\epsilon}_0 . \quad (2.39)$$

The other conservation equation, $\dot{\mathcal{P}} + \mathcal{K}' = 0$, can be used to determine $\mathcal{K}_1$ up to a constant. The latter represents the usual additive ambiguity to the stress tensor, which is set to 0 by requiring the standard thermodynamic interpretation, namely $\langle T_{\sigma\sigma} \rangle = -\frac{\partial E}{\partial R}$, as is done in the derivation of the Burgers equation [3, 4]. Altogether, we find

$$\mathcal{H}_1 = -4 \sum_{m,n} L_m \bar{L}_n e^{i(m-n)\sigma} - \sum_m m^2 \left[ (\alpha_0 + \beta_0) L_m e^{im\sigma} + (\bar{\alpha}_0 - \bar{\beta}_0) \bar{L}_m e^{-im\sigma} \right] ,$$

$$\mathcal{P}_1 = i \sum_m m^2 \left[ \left( \frac{c}{3} - \alpha_0 - \beta_0 \right) L_m e^{im\sigma} - \left( \frac{c}{3} - \bar{\alpha}_0 + \bar{\beta}_0 \right) \bar{L}_m e^{-im\sigma} \right] , \quad (2.40)$$

$$\mathcal{K}_1 = 12 \sum_{m,n} L_m \bar{L}_n e^{i(m-n)\sigma} + \sum_m m^2 \left[ (\alpha_0 + \beta_0) L_m e^{im\sigma} + (\bar{\alpha}_0 - \bar{\beta}_0) \bar{L}_m e^{-im\sigma} \right] .$$

The $T\overline{T}$ trace equation $\operatorname{Tr} T = -2\lambda \det T$ is obeyed at this order,

$$\mathcal{H}_1 + \mathcal{K}_1 = 8 \sum_{m,n} L_m \bar{L}_n e^{i(m-n)\sigma} = -2(\mathcal{H}_0 \mathcal{K}_0 - \mathcal{P}_0^2) . \quad (2.41)$$

It is interesting to note that the $\det T$ operator on the right-hand side of this equation appears without any derivatives at this order. In our formalism, this operator does not necessarily coincide with the deformation added to the Hamiltonian $\mathcal{H}_1$ (2.36) — it does so only for $\alpha_0 + \beta_0 = 0 = \bar{\alpha}_0 - \bar{\beta}_0$. One could choose to impose that these operators coincide also at higher orders, including derivative terms, but we will not do so here.

The undetermined coefficients $\alpha_0 + \beta_0$ and $\bar{\alpha}_0 - \bar{\beta}_0$ do not impact the integrated Hamiltonian and momentum, but due to eq. (2.39) they cannot be chosen in such a way that all derivative terms in the stress tensor vanish. In fact, this ambiguity was to be expected: it represents the freedom to add an improvement term $\Delta T_{ij} = \lambda(\partial_i \partial_j - \delta_{ij} \partial^2) f(T_{kl})$ at first order, with $f = -(\alpha_0 + \beta_0)T - (\bar{\alpha}_0 - \bar{\beta}_0)\overline{T}$.

There were no ordering ambiguities at first order: the only product of operators being proportional to $L_m \bar{L}_n$, which commute. Nevertheless, to consistently go to higher orders, we will consider deforming the theory with a smeared version of the $T\overline{T}$ operator instead. One subtlety that arises is that the conservation equation is not identically satisfied before

taking the regulator away. Rather, using the solution (2.39), the conservation equation eq. (2.38) becomes

$$\frac{c}{3}\sum_m m^3(w_m - 1)w_m(L_m e^{im\sigma} + \bar{L}_m e^{-im\sigma})$$

$$+ 4\sum_{m,n} L_m \bar{L}_n e^{i(m-n)\sigma}(w_m - w_n)[(2m-n)w_m + (m-2n)w_n] = 0 \ . \tag{2.42}$$

The first line is a "smeared local operator" that converges to 0 in the weak sense as $w_n \to 1$. The last line is well-defined as a smeared local operator because the $L_m$ commute with $\bar{L}_n$.

The analog of this at higher orders, where we do find terms with nontrivial commutation relations, is that all singular terms in the OPE cancel before taking the regulator away.[26] We can therefore require conservation at second order up to terms that vanish as $w \to 1$. This procedure yields the following integrated quantities

$$H_2 = 8\sum_{m,n} w_m w_n w_{m+n}(L_m L_n \bar{L}_{m+n} + L_{m+n}\bar{L}_m \bar{L}_n)$$

$$+ \frac{c}{3}\sum_m m^2 w_m^2(L_m L_{-m} + \bar{L}_m \bar{L}_{-m}) + c_1 \sum_m m^2 w_m^2 L_m \bar{L}_m \ ,$$

$$P_2 = 0 \ ,$$

$$K_2 = -40\sum_{m,n} w_m w_n w_{m+n}(L_m L_n \bar{L}_{m+n} + L_{m+n}\bar{L}_m \bar{L}_n)$$

$$- \frac{5c}{3}\sum_m m^2 w_m^2(L_m L_{-m} + \bar{L}_m \bar{L}_{-m}) + c_2 \sum_m m^2 w_m^2 L_m \bar{L}_m \ , \tag{2.44}$$

where $c_1$ and $c_2$ are undetermined constants like those in eq. (2.37).[27] Clearly, $c_1$ is nothing but the constant $g_{1,1}^{(2)}$ identified in section 2.1. The unintegrated stress tensor elements are rather complicated at this order, so we record here only the trace equation after taking

---

[26]The canonical example is $\sum_{m,n}(w_m - w_n)L_m L_n e^{i(m+n)\sigma}$. This is nothing but

$$\int \frac{d\sigma_1}{2\pi} w(\sigma_1)\left[T(\sigma + \sigma_1)T(\sigma) - T(\sigma)T(\sigma - \sigma_1)\right] \ . \tag{2.43}$$

To check whether this is well-defined as an operator, it is sufficient to substitute the universal (divergent) part of the operator product expansion into this expression and observe that all terms cancel.

[27]To be precise, they are

$$c_1 = -\frac{2c}{3} + 6(\alpha_0 + \bar{\alpha}_0 + \beta_0 - \bar{\beta}_0) - (\beta_{T'\overline{T}} - \beta_{T\overline{T}'}) \ ,$$

$$c_2 = \frac{2c}{3} - 18(\alpha_0 + \beta_0) + 6(\bar{\alpha}_0 - \bar{\beta}_0) - 16\gamma_0 - 8\bar{\gamma}_0 + 5(\beta_{T'\overline{T}} - \beta_{T\overline{T}'}) - 2(\gamma_{T'\overline{T}} - \epsilon_{T'\overline{T}}) \ , \tag{2.45}$$

where $\beta_{T'\overline{T}}$ is the coefficient of $T'_w(x) \times \overline{T}_w(x)$ in $\mathcal{B}_1$ and $\beta_{T\overline{T}'}$ is the coefficient of $T_w(x) \times \overline{T}'_w(x)$. The coefficients $\gamma_{T'\overline{T}}$ and $\epsilon_{T'\overline{T}}$ are the coefficients of the former in $\mathcal{C}_1$ and $\mathcal{D}_1$, respectively. At the end of the day, we see that the total derivatives added to $\det T$ which influence the second order Hamiltonian are $\dot{T}'$, $\dot{\overline{T}}'$, $T''$, $\overline{T}''$, $\lambda T'\overline{T}$, and $\lambda T\overline{T}'$.

away the regulator[28]

$$\text{Tr}\,T + 2\lambda \det T|_{\lambda^2} = \tilde{c}_2 \sum_{m,n} mn L_m \bar{L}_n e^{i(m-n)\sigma} \,, \tag{2.47}$$

which is consistent with 0 for some choice of the undetermined constants. However, even if $\tilde{c}_2$ is not set to 0, this is a total derivative $\propto \partial_z \partial_{\bar{z}}(T\overline{T}) + \mathcal{O}(\lambda)$, and can be seen as an improvement term. Its appearance is consistent with the observations in [25, 26] that, at least in the case of $T\overline{T}$-deformed boundary gravitons which was investigated there, a total derivative counterterm has to be added to the $T\overline{T}$ deformation at second order.[29]

## 3 KdV Charges and Integrability

In the previous section we showed that we can perturbatively construct a local Hamiltonian with the energy spectrum given in eq. (1.2). In a CFT, it has been argued that the $T\overline{T}$ deformation preserves the infinite tower of the KdV charges [3, 12]. In turn, this shows that the $T\overline{T}$ deformation preserves integrability.

In our picture this is automatic since the Hamiltonian is unitarily equivalent to a function of the undeformed Hamiltonian and momentum (2.4). In the undeformed theory, the KdV charges are an infinite collection of conserved charges satisfying

$$[I_0^{(i)}, I_0^{(j)}] = 0, \quad [\bar{I}_0^{(i)}, I_0^{(j)}] = 0, \quad [\bar{I}_0^{(i)}, \bar{I}_0^{(j)}] = 0 \,. \tag{3.1}$$

The lowest level ones are $I_0^{(1)} = \frac{1}{2}(H_0 + P_0)$ and $\bar{I}_0^{(1)} = \frac{1}{2}(H_0 - P_0)$. In the deformed theory, one can define analogous charges $\hat{I}^{(i)} \equiv e^{-\lambda X} I_0^{(i)} e^{\lambda X}$ that are automatically conserved. These charges automatically commute with each other and also with the undeformed Hamiltonian, which is a function of $I_0^{(1)}$ and $\bar{I}_0^{(1)}$. However, such charges are not necessary local, in the sense that they are not integrals of local operators. The goal of this section is to argue that there exist combinations of these charges which are local.

In analogy with the Hamiltonian in section 2, we analyze whether there are combinations[30]

$$\tilde{I}_\lambda^{(k)} = \tilde{I}_\lambda^{(k)}(I_i^{(0)}) \,, \tag{3.2}$$

such that

$$I_\lambda^{(i)} = e^{-\lambda X} \tilde{I}_0^{(i)} e^{\lambda X} \,. \tag{3.3}$$

---

[28]The coefficient is given by

$$\tilde{c}_2 = -\frac{8c}{3} - 8(\alpha_0 + \beta_0) + 16(\bar{\alpha}_0 - \bar{\beta}_0) - 16\gamma_0 - 8\bar{\gamma}_0 - 4(\beta_{T'\overline{T}} - \beta_{T\overline{T}'}) - 2(\gamma_{T'\overline{T}} + \epsilon_{T'\overline{T}}) \,. \tag{2.46}$$

[29]The counterterm added in [25, 26], using dimensional regularization, is $\Delta \text{Tr} T = -\frac{\lambda^2 c_0^2}{288\pi^2 \epsilon} f''' \bar{f}'''$. The stress tensor in that theory is $T = -\frac{c_0}{12}(f'^2 - f'')$, so this term does not precisely coincide with what we find here. It would be interesting to investigate if one could instead add only (multiples and derivatives of) the stress tensor as counterterms in the theory of $T\overline{T}$-deformed boundary gravitons. If so, one could expect the leading term to be of the form (2.47).

[30]Note again the difference between the "conjugated charges" $\hat{I}^k$ defined in the previous paragraph and the "fake KdV charges" $\tilde{I}^{(k)}$ used here.

is local. Importantly, such combinations are automatically conserved since they commute with the deformed Hamiltonian $H_\lambda = I_\lambda^{(1)} + \bar{I}_\lambda^{(1)}$

$$[I_\lambda^{(i)}, H_\lambda] = e^{-\lambda X}[\tilde{I}_0^{(i)}, \tilde{H}_\lambda]e^{\lambda X} = 0 \ . \tag{3.4}$$

To explore the locality of the above charges, consider the operators

$$i_k = \sum_{n_1 + \cdots + n_{k+1} = 0} L_{n_1} \dots L_{n_{k+1}}, \quad k \in \mathbb{N} \ . \tag{3.5}$$

The undeformed KdV charges are combinations of the above operators which are designed to ensure commutativity and also finiteness. For simplicity we omit the regulator dependence. We will show that, to first order in $\lambda$, there is a function

$$\tilde{I}_\lambda^{(k)} = \tilde{I}_\lambda^{(k)}(i_1, i_2, \dots, i_k) \ , \tag{3.6}$$

such that the operator $e^{-\lambda X}\tilde{I}_\lambda^{(k)}e^{\lambda X}$ is a local charge. We have

$$e^{-\lambda X}i_k e^{\lambda X} = i_k - \lambda[X_0, i_k] + \mathcal{O}(\lambda^2) \ , \tag{3.7}$$

where $X_0$ is given in (2.14). It is straightforward to evaluate the commutator on the right-hand side

$$[X_0, i_k] = 2(k+1)\sum_{n_1 + \cdots + n_{k+1} = \bar{n}_1} L_{n_1} \dots L_{n_{k+}}\bar{L}_{\bar{n}_1} \tag{3.8}$$

$$+ \frac{(k+1)c}{6}\sum_{n_1 + \cdots + n_k = \bar{n}_1} \bar{n}_1^2 \, L_{n_1} \dots L_{n_k}\bar{L}_{\bar{n}_1} - 2(k+1)\bar{L}_0 \, i_k \ . \tag{3.9}$$

The first two terms are local while the last term is a product of two local charges and hence non-local. However, we can define the following combination

$$\tilde{i}_\lambda^{(k)} = i_k + 2(k+1)\lambda\bar{i}_0 \, i_k + \mathcal{O}(\lambda^2) \ , \tag{3.10}$$

such that $e^{-\lambda X} \, \tilde{i}_\lambda^{(k)} \, e^{\lambda X}$ is a local charge. Hence, we can also define local $\tilde{I}_\lambda^{(k)}$ since these are combinations of $\tilde{i}_\lambda^{(k)}$.

We can compare this result with the one obtained in [42][31], where it was showed that, semi-classically and for $c = 0$, the physical KdV charges satisfy

$$\partial_\lambda I_\lambda^{(k)} = [X_\lambda', I^{(k)}] + 2(k+1)\bar{L}_0 I_\lambda^{(k)} \ , \tag{3.11}$$

for some non-local $X_\lambda'$ which coincides with our $X_\lambda$ at $\lambda = 0$. Taking the expectation of the above equation we get

$$\left\langle \partial_\lambda I_\lambda^{(k)} \right\rangle = 2(k+1)\left\langle \bar{L}_0 I_\lambda^{(k)} \right\rangle \ , \tag{3.12}$$

which exactly matches (3.10) and with the result of [12].

Based on the relation with the semi-classical arguments made in the literature and on the result we obtained for $I^{(0)}$ and $\bar{I}^{(0)}$, we expect that this method will permit to define (quasi)-local KdV charges to all orders in $\lambda$. It would be interesting to explore this further.

---

[31]In our notation, the label $k$ takes all the values in $\mathbb{N}$, while in [42] it takes only odd values.

## 4 Generalized Deformations

In the previous section, we showed that the $T\overline{T}$ deformation has the special property that its fake Hamiltonian, determined by its spectrum, is unitarily equivalent to a (quasi)-local Hamiltonian. In this section, we analyze more general deformations with this property. We study which functions of the conserved charges of the free theory

$$\tilde{H}_\lambda = \tilde{H}_\lambda(I_0^{(k)}) \; , \tag{4.1}$$

are unitary equivalent to a (quasi)-local Hamiltonian $H_\lambda$, in the sense that there is a (non-local) operator $X$ such that

$$H_\lambda = e^{-\lambda X} \tilde{H}_\lambda e^{\lambda X} \; . \tag{4.2}$$

This kind of generalized deformations are interesting because their spectrum is given by the spectrum of the undeformed theory similar to the $T\overline{T}$ case (see (1.2)). As we showed in section 3, such deformations automatically preserve the infinite tower of KdV charges and therefore integrability, though this tower is not guaranteed to consist of local operators. In this section, we only analyze the locality of the Hamiltonian. It would be interesting to analyze the locality of the KdV charges as well, but we leave this for future work.

Let us begin with the most general expression up to order three in the undeformed Virasoro generators

$$\begin{aligned}
\tilde{H}_\lambda =& L_0 + \bar{L}_0 \\
&+ \lambda(b_1 L_0^2 + b_2 \bar{L}_0^2 + b_3 L_0 \bar{L}_0 + b_4 I_0^{(2)} + b_5 \bar{I}_0^{(2)}) \\
&+ \lambda^2 \Big( c_1 L_0^3 + c_2 L_0^2 \bar{L}_0 + c_3 L_0 \bar{L}_0^2 + c_4 \bar{L}_0^3 + c_5 L_0 I_0^{(2)} + c_6 \bar{L}_0 I_0^{(2)} \\
&\quad + c_7 L_0 \bar{I}_0^{(2)} + c_8 \bar{L}_0 \bar{I}_0^{(2)} + c_9 I_0^{(3)} + c_{10} \bar{I}_0^{(3)} \Big) + \dots
\end{aligned} \tag{4.3}$$

The second KdV charge is given by

$$I_0^{(2)} = \sum_{n \in \mathbb{Z}} L_{-n} L_n w_n^2 + a_1 L_0 + a_2 \; , \tag{4.4}$$

where we have used an arbitrary regulator $w_n$ to regulate the infinite sum, as in section 2.1. We have also inserted two regulator dependent constants $a_i$. Requiring finiteness of the expectation value within in a primary state when the regulator is taken away, we have that

$$a_1 = -2 \sum_{n>0} n w_n^2 \; , \tag{4.5}$$

$$a_2 = -\frac{c}{12} \sum_{n>0} n^3 w_n^2 \; . \tag{4.6}$$

The next KdV charge, $I_0^{(3)}$ is complicated, but ultimately local. Hence at second order in (4.2) its presence will not impose any constraints – as such we have omitted the expression

but, in a normal ordering prescription, an expression can be found in [45]. We assume that the momentum does not change,

$$P_\lambda = P_0 = i(L_0 - \bar{L}_0). \tag{4.7}$$

Notice also that we have allowed terms with non-zero spin, which break Lorentz symmetry.

To fix the coefficients in (4.3) that lead to a local first-order Hamiltonian, we look for an $X_0$ such that

$$H_1 = \tilde{H}_1 + [\tilde{H}_0, X_0] . \tag{4.8}$$

where $\tilde{H}_0 = L_0 + \bar{L}_0$. We immediately conclude that the terms proportional to $b_4$ and $b_5$ are local since both $I_0^{(2)}$ and $\bar{I}_0^{(2)}$ are integrals of the local currents. On the other hand, the term $L_0^2$ is not local and it can not be produced by a commutator of $L_0 + \bar{L}_0$. Hence, $b_1 = 0$. Similarly, $b_2 = 0$. We conclude that the most general Hamiltonian to first order is

$$\tilde{H}_\lambda = L_0 + \bar{L}_0 + \lambda \left( b_3 L_0 \bar{L}_0 + b_4 I_0^{(2)} + b_5 \bar{I}_0^{(2)} \right) + \dots$$

with

$$H_\lambda = L_0 + \bar{L}_0 + \lambda \left( b_3 \sum_n L_n \bar{L}_n w_n + b_4 I_0^{(2)} + b_5 \bar{I}_0^{(2)} \right) + \dots$$

$$X_\lambda = -\frac{b_3}{2} \sum_{n \neq 0} \frac{1}{n} L_n \bar{L}_n w_n + \dots . \tag{4.9}$$

Following the same steps, to next order we can conclude that $c_1 = c_4 = c_5 = c_8 = 0$ while the $I^{(3)}$ and $\bar{I}^{(3)}$ are automatically local. The remaining coefficients $c_2, c_3, c_6$ and $c_7$ can be fixed by requiring the third equation in (2.6). One finds

$$c_2 = \frac{b_3^2}{2}, \quad c_3 = \frac{b_3^2}{2}, \quad c_6 = 0, \quad c_7 = 0 . \tag{4.10}$$

We were led to the above by assuming that the regulator dependent constants in (4.4) are non-zero. In conclusion, we have a family of Hamiltonians parameterized by 5 constants $b_3, b_4, b_5, c_9$ and $c_{10}$. To this order, all the parameters except $b_3$ correspond to choosing a KdV current as a Hamiltonian. The remaining coefficient $b_3$ corresponds to the $T\bar{T}$ deformation. It would be interesting to classify all possible generalized Hamiltonians at higher orders, including the first non-trivial $I\bar{I}$ deformation studied in [3] which appears at order $\lambda^3$, but we defer this investigation to future work.

## Acknowledgments

We are grateful to Monica Guica and Per Kraus for comments on the draft. We also thank Seolhwa Kim, Kyriakos Papadodimas, and Alexander Zhiboedov for useful discussions. KR is supported by the Simons Collaboration on Global Categorical Symmetries and also by the NSF grant PHY-2112699. RMM thanks the CERN theory department for hospitality during a portion of this work.

## A    Regulation by smearing

Although Zamolodchikov's argument guarantees that certain observables, like the energy spectrum and S-matrix elements, will turn out finite, we need regulate several of the intermediate expressions to keep control over the calculation. This also grants us access to more fine-grained observables, which are not covered by the original arguments.

To illustrate the ambiguities that appear otherwise, we can consider sums like the ones that appear in the square of the stress tensor, $\sum_{m,n} e^{i(m+n)\sigma} L_m L_n$. As was already pointed out in [42], performing a seemingly innocuous relabeling of indices $m \leftrightarrow n$ and using the Virasoro algebra, this term becomes

$$\sum_{m,n} e^{i(m+n)\sigma} \left( L_m L_n + (n-m) L_{m+n} + \frac{c}{12} n^3 \delta_{m+n} \right) \ , \tag{A.1}$$

which differs from the original by a formally divergent, operator-valued sum.

One physically intuitive way to regulate the theory is by smearing each stress tensor operator with a window function

$$T(\sigma) \to T_w(\sigma) \equiv \int dy\, w(y) T(\sigma - y) = \sum_m w_m L_m e^{im\sigma} \ , \tag{A.2}$$

where $w_m$ are the Fourier coefficients of $w(\sigma)$. At the end of the calculation we take away the regulator by sending $w(x) \to \delta(x)$ or, equivalently, $w_n \to 1$. In order to keep $T$ Hermitian, we require $w_{-n} = w_n^*$. Each term in the result (A.1) gets multiplied by $w_m w_n$, which makes the last two terms

$$\sum_{m,n} (n-m) w_m w_n L_{m+n} e^{i(m+n)\sigma} \ , \quad \sum_m \frac{c}{12} m^3 w_m w_{-m} \ , \tag{A.3}$$

respectively. The latter is finite as long as we require $w_m$ to go to zero faster than $|m|^{-3/2}$ at large $|m|$,[32] and the former is a finite sum of local operators. If we take a symmetric smearing function, with $w_{-m} = w_m$, these vanish identically as they are antisymmetric under $m \leftrightarrow n$ and $m \to -m$ respectively. We conclude that the smeared product $T_w(\sigma) T_w(\sigma)$ is defined unambiguously.

An alternative approach is to define a smeared product, which we can use to regulate the deforming operator $\det T$. At first order it becomes

$$T(\sigma) \times_w \overline{T}(\sigma) \equiv \int dy\, w(y) T(\sigma + y) \overline{T}(\sigma - y) = \sum_{m,n} w_{m+n} L_m \bar{L}_n e^{i(m-n)\sigma} \ , \tag{A.4}$$

With the analogous smearing for $T(\sigma)^2$, the problematic terms in (A.1) now become

$$\sum_{m,n} e^{i(m+n)\sigma} (n-m) w_{m-n} L_{m+n} + \frac{c}{12} \sum_m m^3 w_{2m} \ . \tag{A.5}$$

Again imposing $w_{-n} = w_n$ and appropriate fall-off conditions for $w$, this sum vanishes as an operator due to (anti)symmetry in the summation indices.

---

[32]In what follows, we will require it to fall off faster than any power of $m$.

When defining composite operators, there is a certain amount of freedom in how we regulate. The smearing definitions given above have natural implications for locality, so they make the most sense for us to consider in the present work, but they are not unique. Indeed, we could define $T^3(\sigma)$ by $T_w(\sigma)^3$ just as well as

$$T_w(\sigma)^2 \times_w T(\sigma). \tag{A.6}$$

Of course, in the regulated theory these are distinct operators that will have different matrix elements, but both are valid regulations of $T^3(\sigma)$. The fact that a given composite operator can be regulated in distinct ways corresponds to the familiar ambiguity in finite part when one subtracts a divergence.

With this ambiguity in mind, in this work we use regulators that are motivated by the smearings described above, but will not attempt to regulate all operators in exactly one of these two smearings. Rather, we will prefer to regulate each operator that appears in our Hamiltonian in a manner that happens to be most convenient for our purposes.

## B  Zamolodchikov's argument in perturbation theory

As it is important for the argument in the main text that there exists a well-defined "$T\overline{T}$" operator at each order in perturbation theory, we will dedicate this appendix to justifying this in some detail. We closely follow the original arguments in [1] but point out some tacit assumptions and show how it remains valid when the original theory gets $T\overline{T}$ deformed.

The starting point is a theory which, in the UV, tends to a unitary 2d Euclidean CFT that has Hilbert space spanned by normalizable states on circles corresponding to the quasi-primary and $SL(2,\mathbb{C})$ descendant operators in the theory, which include the conserved stress tensor. The theory then has a convergent OPE [46]. Following [1], we use this to write

$$T(z,\bar{z})\overline{T}(0,0) - \Theta(z,\bar{z})\Theta(0,0) = \sum_a C_a(z,\bar{z})O_a(0,0) , \tag{B.1}$$

where the sum $a$ runs over the aforementioned set of operators $O_a$ associated to the basis of normalized states $|a\rangle$ that span the Hilbert space. We can split them up into quasi-primaries $O_i(0)$ that we label with indices $i, j \ldots$, associated to $|i\rangle \equiv O_i(0)|0\rangle$, and their $SL(2,\mathbb{C})$ descendants $O_\alpha(0)$ with indices $\alpha, \beta \ldots$ and state $|\alpha\rangle = O_\alpha(0)|0\rangle$. The latter contain at least one derivative. These span two orthogonal subspaces.

Using conservation of the stress tensor, reference [1] proceeds to show that

$$\sum_a \partial_z C_a(z,\bar{z})O_a(0,0) = \sum_a \left( c_{T\overline{T}a}(z,\bar{z})\partial_z O_a(0,0) + c_{T\Theta a}(z,\bar{z})\partial_{\bar{z}}O_a(0,0) \right) , \tag{B.2}$$

and similarly for the anti-holomorphic derivative, where $c_{abc}$ are the usual OPE coefficients. We are primarily interested in the terms in eq. (B.1) which are not $SL(2,\mathbb{C})$ descendants. To isolate them, we consider the matrix element of eq. (B.2) in between $\langle i|$ and $|0\rangle$, finding

$$\partial_z C_i = 0 , \tag{B.3}$$

where we used orthogonality of the basis of states to show that the right-hand side, which contains only inner products between the quasi-primary state $\langle i|$ and the $SL(2,\mathbb{C})$ descendent states $\partial_z O_a |0\rangle$ and $\partial_{\bar{z}} O_a(0) |0\rangle$ vanishes. Similarly, we find $\partial_{\bar{z}} C_i = 0$, so eq. (B.1) can be rewritten as

$$T(z, \bar{z})\overline{T}(0,0) - \Theta(z, \bar{z})\Theta(0,0) = \sum_i C_i O_i(0,0) + \sum_\alpha C_\alpha(z, \bar{z}) O_\alpha(0,0) . \qquad \text{(B.4)}$$

The second term on the right-hand side contains only $SL(2,\mathbb{C})$ descendant operators, in other words this is a total derivative contribution that potentially diverges as $|z| \to 0$. The first term is shown to be finite as a result of cluster decomposition [1]. This is the definition of the $T\overline{T}$ operator $\mathcal{O}_{T\overline{T}}(0) \equiv \sum_i C_i O_i(0)$, up to derivatives.

To follow the $T\overline{T}$-flow through theory space, following the strategy in [3, 4], we should make sure that these arguments can be performed at each point along the flow. As already anticipated in those papers, the appearance of shock singularities in the energy spectrum seems to present an obstacle. Here we will argue that the original derivation is at least valid in perturbation theory.[33] As detailed in the main text, at $n$th order in perturbation theory, the stress tensor $T_{\mu\nu}^{(n)}$ is an order $n$ polynomial of the undeformed stress tensor operators $T_{\mu\nu}^{(0)}$ which satisfies the same conservation equations which were essential to obtain eq. (B.2). Similarly, the states are expressed as polynomials of their undeformed counterparts. In other words, perturbation theory takes place within the original Hilbert space of the theory. This observation, together with stress tensor conservation, means that the argument leading to eq. (B.4) goes through unaltered. In particular, and crucially, it uses the undeformed Hilbert space inner product. This last property does not seem to be guaranteed beyond perturbation theory.

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
