# Peer review of "Locality and Conserved Charges in $T\overline{T}$-Deformed CFTs"

_SciPost Physics_

## Round 1 · Referee Report · Christian Ferko (Referee 1) · 2025-3-1

Strengths

(1) The analysis is extremely systematic, clearly laying out its assumptions and proceeding in a logical manner.

(2) Many cross-checks are performed to verify the consistency of the results.

(3) The paper investigates an important open question in the study of $T \overline{T}$ deformations, and nicely connects with many previous observations ($T \overline{T}$ flow of states, etc.).

Weaknesses

(1) Because the work relies on perturbation theory, it is blind to potential non-perturbative effects.

(2) There is not much discussion of the interpretation of the $\mathcal{O} ( c )$ corrections. These terms should be especially important in holographic approaches to the $T \overline{T}$ deformation, which involve a large-$c$ limit.

Report

In this paper, the authors study the $T \overline{T}$ deformation of two-dimensional quantum field theories -- restricting to the special case where the seed theory is conformal -- via perturbation theory in the flow parameter $\lambda$, up to third order. In particular, this work helps to reconcile two different perspectives on the $T \overline{T}$ flow:

(A) the classical definition of the deformation, given in terms of a flow equation for the Lagrangian (or equivalently, the Hamiltonian); and

(B) exact quantum results for quantities in the $T \overline{T}$ deformed theory, such as the inviscid Burgers' equation which describes the flow of the spectrum.

As one might expect, the classical flow (A) provides useful hints about the full behavior of the deformed theory at the quantum level, but does not contain all of the quantum data. One of the main results of this article is that, again up to third order in perturbation theory, one can find a family of local Hamiltonians that is compatible with an ordering of the Hamiltonian which solves the classical flow equation (A) (given by the first term in equation (1.6) of this manuscript), up to correction terms which depend on the central charge. To the best of my knowledge, these interesting $c$-dependent corrections have not been considered before, and they seem essential for reproducing the quantum behavior of the $T \overline{T}$ deformed theory.

The paper is well-written and the analysis is very systematic and methodical. The authors proceed by parameterizing the most general expressions for several deformed quantities up to third order in $\lambda$, which feature undetermined constants due to various ambiguities, and then fix the free parameters using consistency conditions such as conservation of the stress tensor and the expectation that the (quantized) momentum $P$ cannot flow with $\lambda$. Numerous consistency checks are carried out: for instance, the result of parameterizing and then fixing the allowed form of the integrated Hamiltonian (in section 2.1) agrees with the result of performing this same procedure for the stress tensor *densities* (in section 2.3). Furthermore, both of these results agree with the deformed energy spectrum up to the relevant order in perturbation theory, and with the trace flow equation for the stress tensor, for certain choices of the ambiguous quantities.

Because the results of this work are novel and interesting, and address some important open questions about the $T \overline{T}$ deformation, I believe that it merits publication in SciPost Physics. Before proceeding with publication, I have collected a few minor suggestions in the "Requested changes" section of this report, which are not essential but which the authors might consider implementing.

Requested changes

(1) I think the language before equation (1.2), "In particular, (1.1) leads unambiguously to an equation for the energy spectrum [3, 4], the solution of which is given by the square root formula...", could be slightly misleading. This wording makes it sound like the square root formula follows *only* from the flow equation (1.1) and therefore holds for an arbitrary seed QFT. However, the square root formula only applies if the seed theory is a CFT.

It is easy to see this by restricting the inviscid Burgers' equation $\partial_\lambda E_n = E_n \partial_R E_n + \frac{P_n^2}{R}$ to the zero-momentum sector, where it admits the solution $E_n ( R , \lambda ) = E_n ( R + \lambda E_n ( R, \lambda ) , 0 )$. If the seed theory is a CFT, then by dimensional analysis, all energies must scale like $E_n \sim \frac{a_n}{R}$ for dimensionless constants $a_n$, and combining this scaling with the previous expression immediately gives the desired square-root dependence. However, if the seed theory is not a CFT, it may involve some other mass scale $m$ and the energies will then depend on the dimensionless combination $m R$. In such cases, the solution to the algebraic equation above for $E_n ( R , \lambda )$ will not exhibit the same square root functional form.

Of course, in this work the authors always restrict to the case where the seed theory is a CFT, so it is reasonable that they implicitly use this assumption in quoting the solution (1.2). Nonetheless, I might suggest slightly changing the phrasing of this passage to avoid giving the reader the mistaken impression that this solution follows only from (1.1) and not the assumption of a conformal seed.

(2) In footnote 5, the authors point out that the Hamiltonian is not Hermitian for $\lambda < 0$. In fact, even for positive values of the flow parameter, there is an upper bound on the size of $\lambda$ for which the Hamiltonian remains Hermitian, because the ground state energy of a CFT on a cylinder of radius $R$ is $- \frac{c}{12 R}$. Thus the Hamiltonian (1.4) is Hermitian only if the quantity $1 - \frac{\lambda c}{3 R}$ is positive.

(3) The language below equation (1.6), "In the special case where $c = 0$ and the seed CFT is the free boson...", sounds slightly confusing to me. Of course, if the seed CFT is a single free boson, then $c = 1$, so it is not possible to have both $c = 0$ and a free boson CFT seed. I understand this condition to mean "If the seed CFT is the free boson and central charge corrections are ignored...". If this interpretation is correct, maybe the authors can clarify accordingly in a revised submission.

(4) Above equation (2.26), I think the phrase "These modes obey the Virasoro with..." should read "These modes obey the Virasoro algebra with...".

(5) When considering generalized deformations, e.g. in equation (4.3), the authors seem to assume that $\widetilde{H}_\lambda$ is an analytic function of the Virasoro modes and KdV charges in order to write a general series expansion. However, one could (at least formally) also consider non-analytic combinations like $\sqrt{ L_0 \bar{L}_0 }$, defining its action by $\sqrt{ L_0 \bar{L}_0 } | h, \bar{h} \rangle = \sqrt{ h \bar{h} } | h, \bar{h} \rangle$. Including non-analytic combinations could shed light on whether the root-$T \overline{T}$ deformation might admit a "fake" Hamiltonian which is unitarily equivalent to a quasi-local Hamiltonian. Perhaps the authors could comment on this possibility.

Recommendation

Ask for minor revision

  • validity: high
  • significance: good
  • originality: good
  • clarity: high
  • formatting: excellent
  • grammar: excellent

Author:  Ruben Monten  on 2025-06-02  [id 5536]

(in reply to Report 1 by Christian Ferko on 2025-03-01)

We would like to thank Christian for his careful review and thoughtful comments.

We agree with comments 1 through 4.
(1) The square root formula for the energy spectrum is indeed specific to deformed CFTs, which is the case we are restricting to in this paper. We have clarified the wording around equation (1.1).
(2) We have also corrected footnote 5.
(3) We have changed the discussion around equation (1.6) in the way the referee suggested.
(4) Thank you for noticing the typo.

As for the last comment (5), we have clarified the language in our paper to make it clear we only consider fake Hamiltonians that are polynomial in the undeformed charges. Going beyond this poses a nontrivial challenge in our formalism, because it is not clear that one can unambiguously separate local and non-local terms when starting from a non-analytic fake Hamiltonian. To take the example of the root-$T \overline T$ deformation,
$$ \int d\sigma \sqrt{T \overline T} = \int d\sigma \sqrt{ \sum_{m, n} e^{i (m-n) \sigma} L_m \overline L_n} $$
does not separate into a sum of terms of which $\sqrt{L_0 \overline L_0}$ is one. We appreciate the referees point that one can define the operator $\sqrt{L_0 \bar L_0} |h, \bar h\rangle = \sqrt{h \bar h} | h, \bar h \rangle$ on primary states, but its putative completion into a local charge like
$$ \sum_m \sqrt{L_m \overline L_m} = \sum_{m, n} \int d \sigma e^{i (m - n) \sigma} \sqrt{L_m \overline L_n} $$
would instead seem to contain the square root of the stress tensor itself. Although we are not certain this can be done consistently, we think addressing this question is beyond the scope of the current project.

Attachment:

LocalityAndConservedChargesInTTbar-DeformedCFTs_v2.pdf

---

## Round 1 · Referee Report · Anonymous (Referee 2) · 2025-3-3

Strengths

1- The paper provides a careful perturbative analysis of TTbar-deformed CFTs, constructing Hamiltonians up to the third order in the perturbing parameter while ensuring consistency with the flow equation and with quasi-locality of the theory.

2-The study establishes a concrete framework for examining the quasi-locality of TTbar-deformed theories and the conservation of KdV charges. Identifying new central charge-dependent corrections broadens the understanding of quantum effects in the deformed theory.

Weaknesses

1- The approach is perturbative, implying that the analysis is restricted to the Hilbert space of the seed CFT.

2- There is no discussion relating these outcomes to the earlier CFT perturbation theory results.

Report

The paper studies the structure of TTbar-deformed two-dimensional conformal field theories by constructing the deformed Hamiltonian within the perturbation theory framework.
The authors systematically derive a quasi-local Hamiltonian that retains key features of the undeformed theory while introducing new corrections that depend on the central charge.
An important outcome of this work is the explicit verification that a unitary transformation can map the so-called "fake Hamiltonian" to a local one, thereby resolving previous ambiguities in defining the deformed energy spectrum.

One of the strongest aspects of this work is the treatment of KdV charges.
The authors demonstrate that the deformed theory maintains an infinite tower of conserved charges, offering a new perspective on the integrability properties of TTbar-deformed systems.
The detailed consideration of locality constraints, particularly through the analysis of total derivative terms in the stress tensor, further reinforces the findings.

However, certain aspects would benefit from additional discussion or clarification.
Although the paper focuses on establishing theoretical consistency, it does not address potential non-perturbative effects.

Additionally, while the authors acknowledge the ambiguity in selecting the deformed Hamiltonian, further exploration of its physical implications would be valuable.

Given the high quality of the analysis, the importance of the results, and the relevance of the research framework, I consider this paper suitable for publication, provided that at least the following minor points are addressed.

Requested changes

1- It would be nice to see a comparative discussion with the early results of TTbar perturbation theory, particular with:

-A. B. Zamolodchikov, Nucl. Phys. B 358 (1991) 524. -T. R. Klassen and E. Melzer, Nucl. Phys. B 370 (1992), 511-550.

2-In the abstract, the sentence "Zamolodchikov energy spectrum" is unclear. The main text references two papers appearing on the same day [3,4] with additional authors. Moreover, for free bosons and generic TTbar-perturbed CFTs, the spectrum appeared first in the TBA/TTbar-related earlier papers:

Reference [2] for free bosons. Reference [7] for generic CFTs.

The authors should clarify this point in the introduction, particularly around equation (1.2). Furthermore, they may want to use a more inclusive or general phrasing in the abstract when referring to this result.

Recommendation

Ask for minor revision

  • validity: high
  • significance: high
  • originality: high
  • clarity: top
  • formatting: excellent
  • grammar: excellent

Author:  Ruben Monten  on 2025-06-02  [id 5537]

(in reply to Report 2 on 2025-03-03)

We would like to thank the referee for their thoughtful review and astute comments.

We addressed comment number 2 by changing the misnomer "Zamolodchikov energy spectrum" in the abstract and properly referencing earlier results that obtained the energy formula.

As the referee rightfully suggests, this equation can be compared to the results by Zamolodchikov and Klassen--Melzer for the energies of the ground state and first excited state of the QFT obtained by flowing from the tri-critical to the critical Ising model, expanded near the IR.

The ground state energy is given in equation (3.17) in Zamolodchikov and (4.18) in KM. Comparing at linear order allows us to identify our deformation parameter $\lambda$ with $12 t = 8\pi / (M R)^2$. We then find that the $T \overline T$ flow of the ground state with $E_0 = -1/24$ is contained within the old results: it corresponds to dropping all factors of $\pi$ in (3.17) of Zamolodchikov and (4.18) of KM, which we interpret as isolating the contribution of the $T \overline T$ operator from those of the other irrelevant operators in the flow.

The same goes for the first excited state, given in (4.17) of KM. Dropping all factors of $\pi$ in this equation exactly reproduces the $T \overline T$ flow of the state with undeformed energy $E_0 = 1/12$.

We have added a footnote to this extent to equation (1.2) in our paper.

Attachment:

LocalityAndConservedChargesInTTbar-DeformedCFTs_v2_5Chbqto.pdf

---

## Editorial Decision

resubmitted